# Absence of Missense Variant Detection in Inherited Dysfibrinogenemia May Result from a Poor Raw Data Analysis Algorithm or Mosaicism

**DOI:** 10.3390/ijms242316551

**Published:** 2023-11-21

**Authors:** Philippe De Mazancourt, Elisabeth Mazoyer, Myriam Hormi, Michel Hanss

**Affiliations:** 1UMR1179, Université de Versailles-Saint-Quentin, 1 Rue de la Source de la Bièvre, 78180 Montigny le Bretonneux, France; 2Laboratoire de Biologie Moléculaire, Hôpital A. Paré, GHU APHP Paris-Saclay, 9 Avenue Charles de Gaulle, 92100 Boulogne-Billancourt, France; 3Département d’Hématologie, Hôpital Européen Georges Pompidou, GHU AP-HP Centre—Université Paris Cité, 20 Rue Leblanc, 75015 Paris, France; 4Service d’Hématologie Biologique, GHU APHP Paris-Seine-St-Denis, Site Avicenne, 125 Rue de Stalingrad, 93000 Bobigny, France; elmazoyer@gmail.com (E.M.); myriam.hormi@aphp.fr (M.H.); 5Laboratoire d’Hématologie, Centre de Biologie et Pathologie Est, CHU de Lyon HCL—GH Est, 59 Boulevard Pinel, 69677 Bron, France; michel.hanss@chu-lyon.fr

**Keywords:** dysfibrinogenemia, mosaicism, bioinformatic pipeline

## Abstract

Variant identification underlying inherited dysfibrinogenemia quite exceptionally fails. We report on two dysfibrinogenemia cases whose underlying DNA variant could not be identified by Sanger analysis. These failures result from two distinct mechanisms. The first case involved raw signal overcorrection by a built-in software, and the second constituted the first description of mosaicism for one of the fibrinogen genes. This mosaicism was subsequently identified by next-generation sequencing reanalysis of the sample.

## 1. Introduction

Dysfibrinogenemia (OMIM #616004) is a clearcut biological condition defined by a discrepancy between functional [Ac] and quantitative [Ag] assays ([Ag]/[Ac] ratio > 2). Variations in two residues, namely fibrinogen A-alpha chain precursor p.Arg35 and fibrinogen gamma chain precursor p.Arg301, corresponding to Arg16 and Arg275 on the blood protein, respectively, account for more than 70% of dysfibrinogenemias [1]. Most of the remaining variants are missense or small in-frame indels located in domains coded by exon 2 of the *FGA* gene and exons 7 to 9 of the *FGG* gene. At the gene level, cases of unsolved inherited dysfibrinogenemia are quite exceptional. We report on two different dysfibrinogenemia cases, which required a second assessment before being resolved, and highlight distinct bioinformatics limitations.

## 2. Materials and Methods

Functional fibrinogen assays were performed on citrated plasma according to the von Clauss method with Dade^®^ Thrombin reagents (Siemens, Munich, Germany). Quantitative fibrinogen assays were performed on citrated plasma by quantitative radial immunodiffusion (NOR-Partigen^®^ Fibrinogen, Siemens) or with an immunonephelometric assay (Turbitimer, Dade Behring, Paris, France).

For Sanger sequencing with the MegaBACE sequence analyzer (Amersham Bioscience, Piscataway, NJ, USA), the dideoxy sequencing of amplified fragments was carried out using the ET-terminator kit (Amersham Bioscience), and the analysis was performed with the built-in MegaBACE software Cimarron (V4.00 and V3.12).

For Sanger sequencing with the ABI3500 xlDx (Applied Biosystems, Courtaboeuf, France), the dideoxy sequencing of amplified fragments was performed using BigDye Terminator reagents (v.1.1 cycle sequencing RR-100; Applied Biosystems). Bioinformatics analysis was performed with the SeqScape software V 3.0 (Applied Biosystems, ref. 4474978).

Regarding NGS (next-generation sequencing), a capture custom panel including coding sequences of 55 genes and the whole *FGA* (NM_000508), fibrinogen *FGB* (NM_005141), and *FGG* (NM_000509) genes was designed by Twist Bioscience (San Francisco, CA, USA). Captured libraries (Twist Library Preparation Enzymatic Fragmentation Kit, Twist Bioscience) were loaded onto a MiSeq sequence analyzer (Illumina, Evry, France). Gene coverage was above 97% and the coding sequence read depth was >500 for 24 sample runs.

## 3. Results

The first case illustrates raw data signal overcorrection by the onboard software of the capillary electrophoresis system. A 22-year-old patient presented at 30 weeks of her first pregnancy for a biological evaluation, in the context of known relatives with dysfibrinogenemia. She reported no hemorrhagic tendencies or thrombosis events. The biological evaluation revealed normal prothrombin and activated partial thromboplastin times (PT and APTT). The thrombin time (TT) was 43 s (control: 17 s) and dysfibrinogenemia was identified (Clauss activity: 0.8 g/L and normal antigen: 3.5 g/L with the immunonephelometric assay). Dideoxy sequencing of amplified fragments was performed on a MegaBACE sequence analyzer and did not identify the variant in the first place. Reanalysis of the raw data with a different algorithm (a previous version, V3.12, of the built-in software provided by the manufacturer, Cimarron 4.00) showed that the absence of detection was due to background signal overcorrection (Figure 1A,B). The identified variant was one of the most common causes of dysfibrinogenemia, a heterozygous c.901C > T transition on the *FGG* gene, thus predicting p.Arg301Cys on the gamma chain precursor. At that time, the reanalysis of cold cases demonstrated a few occurrences of the artefact that were not restricted to the *FGG* gene. Errors resulted from defects, either in the former or later version of the software. Identification of this overcorrection led us to switch to the Applied Biosystems sequencing kit and capillary electrophoresis apparatus (ABI3130 then ABI3500), which correctly identified the variant.

The next puzzling data concerned a father (patient I.1), son (patient II.1), daughter (patient II.2), and granddaughter (patient III.1), all carrying dysfibrinogenemia, with Clauss values in the 0.4–0.8 g/L range and quantitative assay results in the 3.0–3.5 g/L range.

The dysfibrinogenemia diagnosis was a fortuitous discovery during preoperative testing of a 36-year-old asymptomatic patient in 1986 (patient II.1). Extensive biological evaluation was performed in 2010. The prothrombin time was slightly increased (the ratio was 47%) and the activated partial thromboplastin time (APTT) ratio was normal. TT was 58.2 s (control: 20.6 s), and the fibrinogen activity determined by the Clauss assay was 0.52 g/L (normal range: 2–4 g/L). The quantitative fibrinogen assay result (radial immunodiffusion) was 3.0 g/L and the D-dimer concentration (VIDAS BioMérieux, Marcy l’Étoile, France) was 87 ng/mL (normal range <250 ng/mL). In 2010, a family screening revealed dysfibrinogenemia in his daughter (patient III.1), 56-year-old sister (patient II.2), and 81-year-old father (patient I.1), based on clotting test data.

The biological data of the propositus’ sister (patient II.2) were a prothrombin ratio of 67%, a TT of 63.4 s (control: 17.8 s), a reptilase time (RT) of 100 s (control: 20 s), fibrinogen activity (Clauss) of 0.68 g/L, and a D-dimer concentration of 207 ng/L. Similar results were observed two years later, in 2012. Her medical file showed no evidence of thrombosis or hemorrhagic complications after appendix surgery, three dental extractions, thyroidectomy, two C-sections, tubal ligation, and hysterectomy. All procedures were performed without prophylactic fibrinogen infusion.

The biological data of the propositus’ father (patient I.1) were a prothrombin ratio of 59%, a factor II concentration of 0.82 IU/mL, a factor V concentration of 0.82 IU/mL, a concentration of factors VII + X of 1.20 IU/mL, a normal APTT, fibrinogen activity (Clauss) of 0.85 g/L, a quantitative fibrinogen assay result of 3.5 g/L, a D-dimer concentration of 591 ng/mL, a TT of 51.1 s (control: 17.8 s), and an RT of 96 s (control: 20 s). Careful clinical evaluation revealed no hemorrhagic tendencies or thrombosis history. His medical history included an excisional hemorrhoidectomy, a cholecystectomy, surgical treatment of a left inguinal hernia, and several dental extractions. None of the surgical procedures were performed under prophylactic fibrinogen infusion, and there were no hemorrhagic or thrombosis complications. He had untreated chronic myelomonocytic leukemia. The patient had not undergone a liver or bone marrow transplant.

DNA was available for patients II.1, II.2, and I.1. Sanger sequencing with Applied Biosystems reagents and devices showed that the two children (II.1 and II.2) carried a c.901C > T causal variant on the *FGG* gene. However, the father (I.1) apparently did not carry the variant. Noise signal overcorrection by the software was ruled out. To eliminate a mis-priming bias, two different sets of primers were used for sequencing. Sample error was also ruled out (two different samples were analyzed 6 months apart). The paternity was confirmed based on short tandem repeat data. Once NGS became available, the father’s blood DNA sample displayed mosaicism for the c.901C > T variant of the *FGG* gene (50/609 reads; 8.2%, see Figure 1C,D). Careful reexamination of the Sanger electropherogram demonstrated a low signal that could not be distinguished from the noise signal.

## 4. Discussion

Accurate genotyping relies on high-performance bioinformatics tools and algorithms. Amersham’s capillary electrophoresis device once used for dideoxy sequencing is no longer sold or updated. However, retrospective analysis of unsolved dysfibrinogenemia cases demonstrated that inadequate raw data treatment has been a cause of variant detection failure, as shown in the first case described. This justifies the reanalysis of suspected dysfibrinogenemia cases, even in cases where the gene analysis yielded negative results and the family study did not provide informative data. The Applied Biosystems Seqscape software V3.0 was not affected by noise overcorrection, as a systematic NGS reanalysis of over a hundred unsolved cases in various diseases did not indicate this possibility.

Some disorders are particularly prone to germline mosaicism. Examples of this phenomenon are Duchenne muscular dystrophy [2] or osteogenesis imperfecta type 1 [3]. Approximately one-third of male births with hemophilia B are reported to be sporadic. However, some mothers later proved to be carriers of mosaicism (at least three out of 149 families) [4]. Mosaicism is the mechanism explaining over 10% of cases of patients with sporadic hemophilia A [4,5,6,7,8]. Mosaicism has not previously been reported in dysfibrinogenemia, although over 1200 variants have been published [9], (https://site.geht.org/base-de-donnees-fibrinogene/), URL accessed on 23 October 2023 and [10]. Different mechanisms have been described to explain mosaicism.

Postzygotic de novo mutations during embryonic development result in gene mosaicism. The three main types are somatic, gonadal, and gonosomal mosaicism, which differ in terms of body distribution of the postzygotic variants [11,12]. Gonosomal mosaicism probably represents the most complex type, with postzygotic mutations affecting both gonadal and somatic cells. Consequently, patients carrying this type of mosaicism are at risk of transmitting the mutant allele to their offspring and might develop symptoms of the disease.

As for the two other mechanisms, gonadal mosaicism is irrelevant here. Indeed, postzygotic mutations are restricted to gonadal tissue, with a moderate-to-high risk of transmission of the mutant allele to the offspring as the main clinical consequence, and an absence of variants in the liver tissue. In somatic mosaicism, postzygotic mutations are located in the somatic cells and may be the cause of disease manifestations related to the mutated gene. This mechanism is also irrelevant as offspring are affected.

In the present case, postzygotic mosaicism, i.e., the variant localized to the liver and gametic cell subsets, is most likely to be involved [13]. We found 8% of variants in peripheral leukocyte DNA. However, the proportion of active plasma fibrinogen represented 13% of the antigen level, which is very similar to the values usually reported in other *FGG* c.901C > T classical heterozygous variant carriers. Thus, the liver variant allele fraction is likely closer to the values expected for heterozygous carriers. The significance of mosaic mutations may be underestimated because parental genotyping is not always performed. However, in our clinical experience, only one case of an apparently de novo variation (*FGG* c.766A > C, p.Asn256His) has been observed [14]. Careful reevaluation of the parental genotypes by NGS failed to identify the mechanism, as none of the parents carried a copy of the variant in their blood cells (0/366 and 0/422 reads).

## 5. Conclusions

To our knowledge, this is the first described occurrence of mosaicism in dysfibrinogenemia. Although this is a very rare event, biologists and clinicians should keep in mind that this might be an explanation for undetected variants in inherited dysfibrinogenemia.

## Figures and Tables

**Figure 1 ijms-24-16551-f001:**
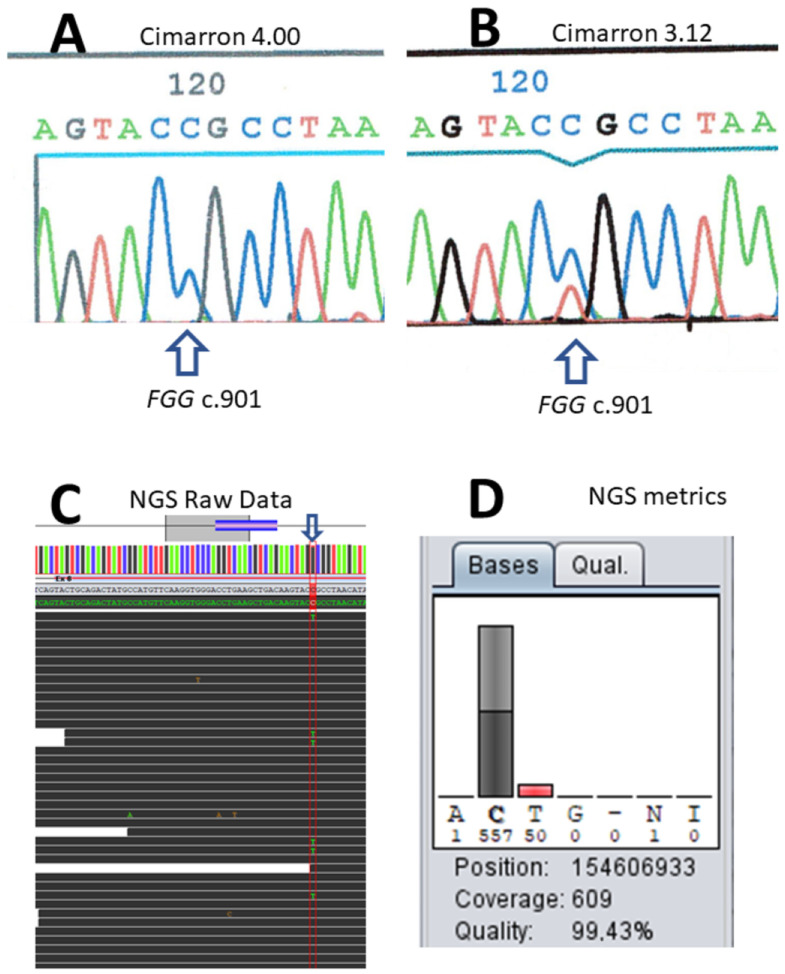
Sanger limits and next-generation sequencing (NGS) analysis of fibrinogen gamma chain (FGG) c.901C > T variant. Figure Legend: The base colors are traditionally labeled so that adenine is green (A), thymine red (T), cytosine is blue (C) and guanine is black. (**A**) Built-in Amersham MegaBACE Cimmaron software (version 4.00) analysis of the raw data from the first example. The vertical arrow points to FGG c.901 (NM_000509), chr4 position 154606933 hg38. (**B**) Cimmaron software (version 3.12) reanalysis of the same raw data. (**C**) Analysis of the raw NGS capture data. Vertical blue arrow points to the mosaicism position. (**D**) Metrics at the c.901 position of the FGG gene, chr4 position 154606933. The nucleotide T represents 50/607 reads.

## Data Availability

The data presented in this study are available on request from the corresponding author. The data are not publicly available to respect anonymity.

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
