# Peer review of "Absence of Missense Variant Detection in Inherited Dysfibrinogenemia May Result from a Poor Raw Data Analysis Algorithm or Mosaicism"

_ijms, 2023, doi:10.3390/ijms242316551_

Round 1
Reviewer 1 Report
Comments and Suggestions for Authors
De Mazancourt et al. present a study where they identified a genetic mosaicism in dysfibrinogenemia. The second described case is very interesting as it is the first report to demonstrate such a phenomenon, and will be of great interest clinically. Can authors provide some description on the clinical presentation of these 3 related patients (farther, son and daughter) about their bleeding phenotype?
There are also a couple of minor edits authors need to make:
1. Line 32 - citation needed.
2. In some parts of the manuscript, the font is different. Please correct.
3. Authors refer to FGG p.Arg301Cys mutation as common, which may prompt a reader to assume that this is the most common mutation, whereas the most common one in dysfibrinogenemia is within FGA.
Comments on the Quality of English LanguageEnglish is nice and clear.
Author Response
We wish to thank you for your interest in our manuscript and both reviewers for their comments which helped improving the manuscript. Of course, we agree with all comments and fixed the following points:
Both reviewers pointed that clinical and biological and clinical descriptions were missing. The available data have been added.
Some comments were added on the possible mosaicism mechanism. All the minor points were fixed.
To make it easier for the reviewers to track the changes, all the modifications have been highlighted in yellow.
Thank you very much for your consideration, yours sincerely, P de Mazancourt
Reviewer 2 Report
Comments and Suggestions for Authors
The Authors report a couple of cases with inherited dysfibrinogenemia and no apparent gene mutations identified by Sanger sequencing Two different meachanisms were indeed identified as possible cause, including a hiterto unreported mosaicism.
Minor comments
-In the title and across the paper, please include (and replace constitutive when appropriate) the term inherited
- A brief indication on clinical aspect of the wo cases is indicated
- Conclusions: please modify: ...biologists and clinicians should keep in mind...; in inherited dysfbrinogenemia
Author Response

(The authors gave the same response as above.)
